

# Familiarity with visual stimuli boosts recency bias in macaques

Nicolas Brunet[1] and Bharathi Jagadeesh[2]

[1] Department of Psychology and Neuroscience, Millsaps College, Jackson, MS, United States of America
[2] Department of Physiology and Biophysics, University of Washington, Seattle, WA, United States of America

## ABSTRACT

To probe how non-human primates (NHPs) decode temporal dynamic stimuli, we used a two-alternative forced choice task (2AFC), where the cue was dynamic: a movie snippet drawn from an animation that transforms one image into another. When the cue was drawn from either the beginning or end of the animation, thus heavily weighted towards one (the target) of both images (the choice pair), then primates performed at high levels of accuracy. For a subset of trials, however, the cue was ambiguous, drawn from the middle of the animation, containing information that could be associated to either image. Those trials, rewarded randomly and independent of choice, offered an opportunity to study the strategy the animals used trying to decode the cue. Despite being ambiguous, the primates exhibited a clear strategy, suggesting they were not aware that reward was given non-differentially. More specifically, they relied more on information provided at the end than at the beginning of those cues, consistent with the recency effect reported by numerous serial position studies. Interestingly and counterintuitively, this effect became stronger for sessions where the primates were already familiar with the stimuli. In other words, despite having rehearsed with the same stimuli in a previous session, the animals relied even more on a decision strategy that did not yield any benefits during a previous session. In the discussion section we speculate on what might cause this behavioral shift towards stronger bias, as well as why this behavior shows similarities with a repetition bias in humans known as the illusory truth effect.

## INTRODUCTION

The cognitive process of selecting among alternatives, hoping for the most favorable outcome, is not unique to humans. Many studies that investigate biases and decision-making strategies in humans have been replicated with non-human animals. Some of those cognitive biases have been observed in various animal species (ranging from pigeons to primates), suggesting a strong evolutionary link. One example is hyperbolic discounting, which reveals a preference for small rewards that occur sooner, rather than larger ones that occur later. See *Vanderveldt, Oliveira & Green (2016)* for a review. Another example is the serial position effect, characterized by the typical U-shaped relationship between an item's position in a list and the probability to recall it; this effect is demonstrated

Corresponding author
Nicolas Brunet,
brunenm@millsaps.edu

for pigeons (*Santiago & Wright, 1984*), rhesus macaques (*Sands & Wright, 1980*), and other species. The hot hand fallacy or the "belief" that experiencing successful outcome leads to a greater chance of success in further attempts was reported in a study with rhesus macaques (*Blanchard, Wilke & Hayden, 2014*). Also the ability to make rational choices, using paradigms that are even challenging for humans, such as the Monty Hall dilemma—a statistical illusion—have been studied in birds (*Herbranson & Schroeder, 2010*), and primates (*Klein et al., 2013*).

Ambiguous stimuli have been proven to be an effective strategy to probe the behavior and cognitive bias in animals. Typically, an animal learns to discriminate between stimuli that predict positive consequences and stimuli that predict a negative outcome. See Roelofs et al. for a comprehensive review (*Roelofs et al., 2016*). Once the animal masters the task, ambiguous stimuli are introduced that lie between the original stimuli. The hypothesis is that the animal's "mood" will bias the choice following an ambiguous test stimulus. If its mood is positive, then it will classify the stimulus as positive. Correspondingly, if the animal's mood is negative, it will classify the stimulus as negative. This approach has been used to show that laboratory rats in unpredictable environments had a more pessimistic attitude than rats in predictable environments (*Harding, Paul & Mendl, 2004*), and that dogs who exhibit high levels of separation-related behavior have a more negative underlying mood (*Mendl et al., 2010*). The approach is now considered as a valuable indicator of animal wellbeing, applicable to many species, ranging from honey bees (*Bateson et al., 2011*) to non-human primates (*Burman et al., 2009*).

In this study, the researchers use ambiguous stimuli, not to test the mood of the animal, but to gain insight about how it visually perceives the ambiguous stimuli. A method often used to measure the subjective experience of primates is the two-alternative forced choice (2AFC) task. Typically, for each trial, an auditory or visual stimulus is presented, which then needs to be categorized into the correct class. The subject does so by selecting one of two possible options. When the answer is correct, animals usually receive food pellets or a juice reward. Similar to the cognitive bias studies to assess mood, described above, we insert "impossible trials" where the cue is ambiguous, once the animal masters the task. The ambiguous cue in this case contains information that can be linked to either image of the choice pair. To mask the unsolvable nature of those trials, ambiguous and unambiguous trials are mixed, and ambiguous trials rewarded at random (50%), regardless of the response.

For this study, we were particularly interested in how familiarity with visual stimuli influences decision and whether this would result in a change of behavioral bias. Familiarity with visual stimuli modulates neural processing along the ventral visual pathway. In humans, visual evoked potentials in response to familiar stimuli are larger than those recorded in response to new stimuli (*Tanaka & Curran, 2001*). Also in rhesus macaques, signals retrieved from electrodes placed over the occipital cortex (*Peissig et al., 2006*) and the temporal lobe (*Anderson et al., 2008*) reveal stronger evoked potentials for familiar compared with novel stimuli. Erickson et al. show that neuronal preferences for pairs of nearby neurons, in inferotemporal cortex, differ greatly for novel stimuli, but become more similar after experience with the stimuli (*Erickson, Jagadeesh & Desimone, 2000*). To probe

whether behavior would be susceptible to the familiarity of the visual stimuli in a task where familiarity with the stimuli (or lack thereof) is irrelevant for performance and reward, we designed a 2AFC task where the cue was a dynamic visual stimulus. Unbeknownst to the animals, a subset of trials, rewarded at random, were ambiguous since there was no correct nor incorrect answer. In order for our approach to work, it was crucial that the animals, in response to the "impossible trials", displayed a non-random behavior consistent with a decision strategy. The results of the first sessions did show that both animals indeed elicited the same biased behavior, suggesting genuine attempts to maximize reward. Satisfied with the preliminary results, we shifted to the main objective of our study: investigate whether reusing the same stimuli (now familiar to the subject) would result in a behavioral shift. The three possible outcomes are the following:

(1) a weakening of the bias, with stimulus familiarity .

This outcome suggests that the animals learned that, for the subset of trials where ambiguous cues are used, reward is unrelated to behavior. Not receiving any benefit from making the mental effort necessary to make a decision might result in random behavior; alternatively, the primates might attempt to minimize efforts by selecting an image based upon its spatial position (for instance, always selecting the image displayed at the bottom), rather than its contents.

(2) a preservation of the bias, with stimulus familiarity .

NHPs responding to familiar stimuli in the same way they do to novel stimuli might indicate that their perception of the stimuli has not been changed. In other words, the mechanisms of how those stimuli are processed temporally are not affected by the familiarity of the stimuli.

(3) a strengthening of the bias, with stimulus familiarity.

This is the most counterintuitive outcome, because the primates are neither rewarded nor trained to display such behavior. A form of long-term adaptation (LTA), potentially associated with learning and/or memory, might take place during the session where the stimuli are novel. If so, then the temporal processing of those stimuli will be modulated when used again during a later session, consistent with the stronger visual evoked potentials observed in human and non-human primates in response to familiar stimuli (*Anderson et al., 2008*; *Peissig et al., 2006*; *Tanaka & Curran, 2001*).

## MATERIALS AND METHODS

### Subjects

Data were obtained from two adult male rhesus monkeys (7–10 kg). Animals were housed in standard cages in the Washington Primate Research Center, with pair housing when possible (*Liu, Murray & Jagadeesh, 2009*). Monkeys B. and S. were trained, acquired, and raised under the same protocol (# 3275-01) as monkeys L. and G. (*Akrami et al., 2008*; *Liu, Murray & Jagadeesh, 2009*). Prior to this study, the animals were well trained in versions of the 2-alternative forced choice design used in this experiment (*Liu & Jagadeesh, 2008*). During each session, NHPs were positioned in a booth, and performed a two alternative forced choice-delayed match to sample task (2AFC). Each trial consisted of a series of

events, displayed on a computer monitor, and ended with a response in the form of a saccade (see Experimental Design). The stimuli were displayed using CORTEX, a program for neural data collection and analysis developed at the NIH (Bethesda, MD). Both NHPs were implanted with a head restraint prosthesis, used to limit movement of the head during data collection, and a scleral eye coil to monitor eye position (DNIm Newarkm DE, USA; (*Judge, Richmond & Chu, 1980*)). To induce motivation for the cognitive task, freely available water was limited the day before the experiment, and fluids (water or juice drops) used as a reward following a correct trial. All animal handling, care and surgical procedures were performed in accordance with guidelines established by the NIH, and approved by the institutional Animal Care and Use Committee (IACUC) at the University of Washington (Protocol number 3275-01).

## Stimuli and experimental design

For each session, one pair of photos was selected from a picture database containing images of everyday objects such as faces, plants, animals, landscapes and buildings (see Fig. 1A for some examples). All visual stimuli used for that session were then derived from those images. All images were 90 × 90 pixels, drawn from a variety of sources, such as databases and personal photo libraries. Stimuli were presented on a computer monitor with 800 × 600 resolution and a refresh rate of 100 Hz. At the viewing distance used, images subtended 4° of visual angle. The visual stimuli were either familiar to the subject (used during a previous session) or novel (never used before).

The sequence for each trial was the same (see Fig. 1C): after the monkey foveated a fixation point for a variable time between 250 and 500 ms within an invisible fixation window that was 4 degrees in diameter and centered around the fixation point, a visual cue was presented. The cue was displayed for 500 ms, and either static or dynamic (see below), and its location and size the same as for the invisible fixation window. After the cue disappeared, the fixation point was still turned on for a variable delay time between 700 and 1,200 ms, and after the choice pair was displayed, for another variable period of 400–700 ms. Only then was the fixation point turned off, signaling the NHP that it was allowed to make a choice. The NHP could express its preference by making a saccade towards either image A or image B (together forming the choice pair). The monkey received a reward for selecting the image (either A or B), that best matched the cue.

To avoid spatial bias, the order of the choice pair, located 5 degrees to the left of the fixation point was randomized: Image A was 5 degrees either up or down, and image B respectively 5 degrees down or up with respect to the fixation point (see Fig. 1C).

For trials where the cue was static (either image A or image B) reward followed the selection of the corresponding image from the choice pair (delayed match-to-sample; see Fig. 1C). For some trials, however, a dynamic cue was used: a movie snippet of 5 frames, drawn from an 11-frame long movie that was generated by morphing image A (frame 1) slowly into image B (frame 11; see Fig. 1B, blue and pink boxes). The images were morphed by using MorphX (http://www.norrkross.com/software/morphx/morphx.php), a program for morphing between two photographic images. The dynamic cues, consisting of either the first or last five frames of the larger movie, were displayed for 500 ms (100 ms per

A Example of image pair

B Example of a motion picture (mp), and the derrived dynamic cues

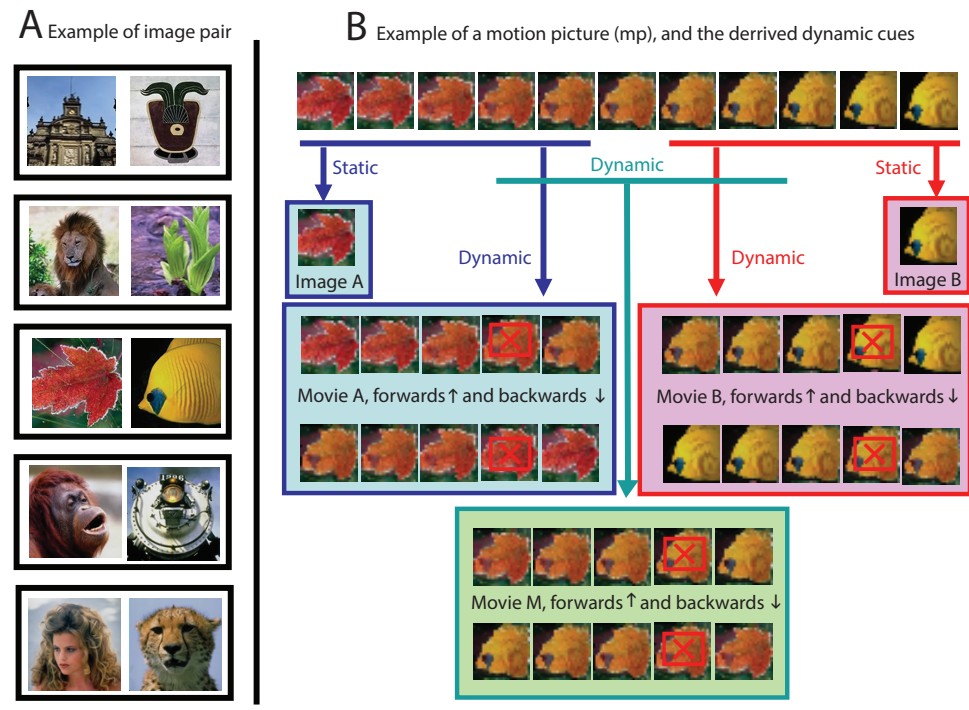

C Order of events, illustrated with example

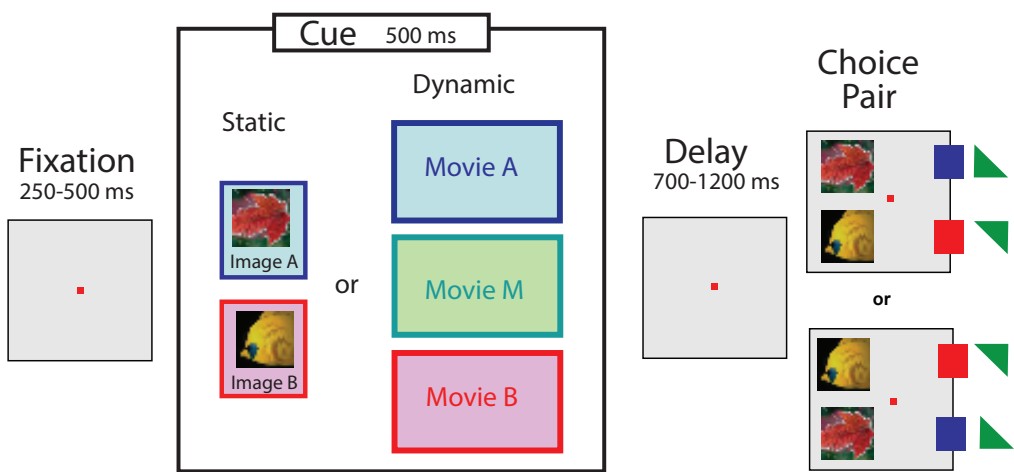

**Figure 1** **Examples illustrating the stimuli and experimental paradigm.** (A) For each session, all stimuli were derived from one pair of images selected from a picture data set (see black boxes for some examples of stimuli pairs). (B) Dynamic cues were drawn from a larger 11-frames animation, generated by morphing both images of an image pair into each other. Static cues were identical to the images of the choice pair (see second row), while dynamic cues were formed by the first five (movie A, blue box) or last five (movie B, pink box) frames of the larger movie. Ambiguous dynamic cues were obtained by selecting the five frames from the middle (movie M, green box). The playing direction of trials with dynamic cue was determined randomly (see both rows in blue, pink and green boxes). Weighted cues were generated by skipping the penultimate frame (see frames marked with a red crossed box), which resulted in a 100 ms gap during the display of the cue. (C) Schematic representation of a trial: First, a cue, either static (image A or image B) or dynamic (movie A, B, or M), was displayed. The color of the boxes in (C) matches the examples shown in (B). Targets, leading to reward, are marked with a small square (100%) or small triangle (50%, randomly determined), and colored to match the cue types shown in the colored boxes.

frame) and presented either in forward or reverse order. Trials with dynamic cues were rewarded when the NHP selected the target (from the choice pair) that was also presented in the cue. More specifically, trials where the cue was a movie snippet consisting of frames 1-2-3-4-5 or 5-4-3-2-1, led to reward when the animal selected image A (equal to frame 1), whereas trials where the cue was a movie snippet consisting of frames 7-8-9-10-11 or 11-10-9-8-7 led to reward when the NHP selected image B (equal to frame 11).

For a subset of the trials, the dynamic cue was ambiguous, consisting out of the five frames drawn from the middle of the 11-frame movie, thus containing neither image A nor image B (see Fig. 1B, green box). The ambiguous cue was presented either in forward (frames 4-5-6-7-8) or reverse direction (frames 8-7-6-5-4). For those ambiguous trials, where there was no correct answer, the animals were rewarded randomly at a 50% chance rate upon completing the trial.

For some of the sessions, the dynamic cue was "weighted". We introduced this condition after we noticed—based upon preliminary results—that the NHPs were more likely to extract information from the end of a dynamic cue than from the beginning (see results). We were interested in whether weighting the movie towards the beginning, by removing the penultimate frame (frames with a red crossed box, in the example shown in Fig. 1B), would change the behavior of the NHPs. The intervention, resulting in a 100 ms long gap during the display of the cue, however made perceptually almost no difference, and consequently did not affect the behavior of the animals (as reported in the result section).

Sessions were organized in blocks of 60 trials. The first block contained only unambiguous static cues (see Fig. 1B, row 2 for an example), which was meant to motivate the subject and to make it familiar with the stimulus pair before confronting it with trails where the cues were dynamic. Subsequent blocks, also 60 trials long, consisted of all 10 conditions, randomly mixed. Those are the two static conditions (Fig. 1B, row 2), the four unambiguous dynamic conditions (Fig. 1B, movie snippets in blue and pink boxes), two ambiguous dynamic conditions (Fig. 1B, green box) always followed by reward, and the same two dynamic conditions, but never followed by reward. We collected minimum five blocks of data for each session, yielding at least 96 trials where ambiguous cues were used. We collected data from 141 sessions for monkey B. and 269 sessions for monkey S., totaling more than 200,000 trials.

## Data analysis

All analyses, including averages, error estimates, and statistical significance were calculated using Matlab, version R2015a; Figures were generated using Matlab, version R2015a, and Adobe Illustrator CS6. Averages are given as unweighted mean $\pm$ SE.

To quantify the results obtained for this study, we used two equations.

The first equation is used to calculate the Preference Index (PI), a metric that quantifies temporal biases which we termed recency and primacy (see results). PI is calculated as follows:

$$PI = \left( \frac{R-P}{R+P} \right) \qquad (1)$$
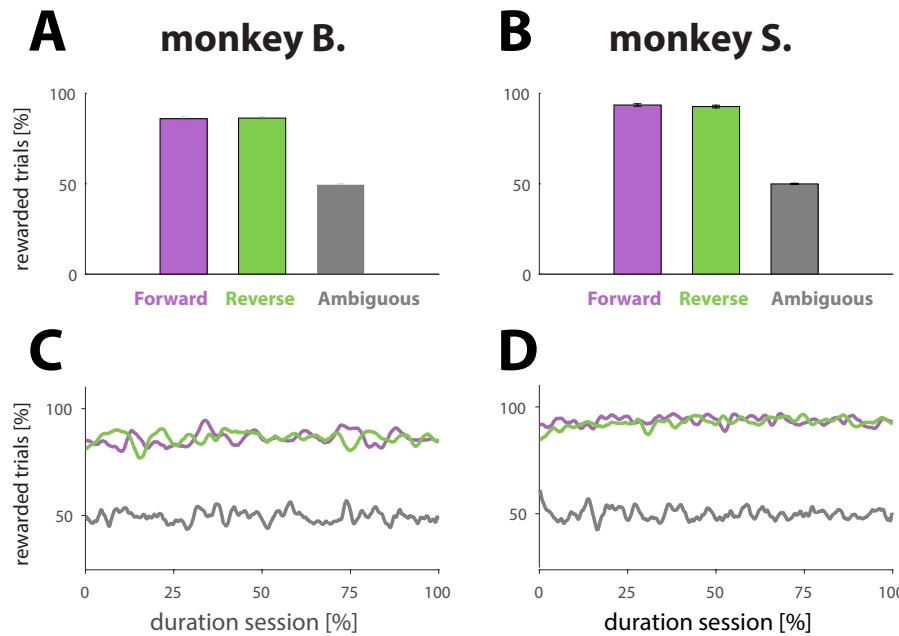

**Figure 2** **Analysis reward ratios for each NHP and each experimental condition.** (A and B) Percentage of rewarded trials for different dynamic cue conditions. Conditions are represented with different colors: purple for unambiguous trials played in the forward direction (for an example, see Fig. 1B, first row in blue box or second row in pink box). Green for unambiguous trials played in the reverse direction (for an example, see Fig. 1B, second row in blue box or first row in pink box). Gray for ambiguous trials in either direction (for examples, see Fig. 1B, green box). The results were computed by averaging performance across all sessions; the error bars denote SE. (C and D) Percentage of rewarded trials as a function of normalized trial number for different conditions (colors match conditions in A and B). The results were obtained by first normalizing each session with respect to trial number and condition, and then averaging across all sessions.

where R and P represent the number of ambiguous trials within a given session where the NHP respectively selected the image most resembling the end (Recency) or beginning (Primacy) of the dynamic cue (see Experimental design and Fig. 2, green box).

A second equation was used to quantify the behavioral shift ($\Delta_{PI}$), between two sessions (S1 and S2) where the same set of visual stimuli were used. We denote S1 as the session where a given set of stimuli was used for the first time and S2 as the session where the same set was used again, thus for a second time. The $\Delta_{PI}$ for a pair of S1 and S2 is obtained as follow:

$$\Delta_{PI} = PI_{S2} - PI_{S2} \tag{2}$$

where $PI_{S1}$ and $P_{IS2}$ are the PI values (see Eq. (1)) calculated for respectively S1 and S2.

## RESULTS

### Discrimination between two visual stimuli, using static and dynamic cues

Before acquisition of the behavior data used for this study, two NHPs were first trained in the 2AFC task using only static cues (Delayed matching-to-sample). Once the NHPs

routinely performed at high levels of accuracy (>90% correct), we also included 5-frame long dynamic cues (see Materials and Methods), drawn from a larger 11-frame long movie generated by morphing image A (frame 1) stepwise into image B (frame 11).

Once the NHPs also mastered trials with dynamic cues (>90% correct), we switched from training to working mode, and composed a complete new set of image pairs, from which we selected a single pair for each session (see Fig. 1A for some example pairs).

The animals performed well on trials with dynamic cues, whether the target frame was included at the beginning (Figs. 2A and 2B, purple bars; forward direction) or end (green bars; reverse direction) of the dynamic cue, as illustrated by the success rates: $86.0 \pm 1.1\%$ and $86.3 \pm 1.0$ (monkey B); and $93.6 \pm 0.7$ and $94.1 \pm 0.6\%$ (monkey S) for respectively forward and reverse playing directions. A two sample $t$-test comparing the means of the success rates computed for each playing direction revealed that the playing direction of the cue had no effect on performance (monkey B.; $p = 0.80$; $N = 107$ and monkey S; $p = 0.40$; $N = 131$). As expected, the NHPs were not able to figure out a strategy allowing them to receive more reward than predicted by chance (Figs. 2A and 2B, gray bars) for trials with ambiguous cues, demonstrating that the reward was truly random: $49.5 \pm 0.5\%$ for monkey B., and $49.9 \pm 0.3\%$ for monkey S. To examine whether performance changed over the course of a session, which might signal fatigue, we also computed the proportion of rewarded trials as a function of trial number for a given condition. To that extent, we normalized each session with respect to the total number of trials. The results, based upon an average of all sessions (Figs. 2C and 2D) suggest that the primates did neither change behavior nor strategy with respect to any of the conditions during the course of a session.

Not displaying the penultimate frame (weighted cues; see Materials and Methods) caused a small interruption in the sequence, and potentially a perceptual shift towards the beginning of the dynamic cue (see Materials and Methods). The intervention, however, did affect neither behavior nor performance as illustrated by a two sample $t$-test comparing the means of the success rates for sessions of each condition: there was no statistical difference between sessions using weighted versus unweighted movies for the forward direction ($p = 0.06$ for monkey B. and $p = 0.14$ for monkey S.), nor for the reverse direction ($p = 0.21$ for monkey B. and $p = 0.76$ for monkey S.).

## Recency and primacy effects

Because the performance of the NHPs for trials with informative cues was very satisfactory, not sensitive to the direction of the dynamic cue, and consistent over the course of a session (see Fig. 2), we were confident that the NHPs, even when facing ambiguous cues, would show non-random behavior by continuing to apply the learned rules for making a "correct" response. To probe the behavior with respect to those "impossible" trials, the ultimate aim of this study, we will from here onwards only consider the data from those trials. Moreover, for this subset of the data, we will no longer focus on correct versus non-correct identification of the target which was determined random and therefore not informative, but instead on the interpretation of the cue by the animal. Would it rely, trying to identify the "correct" target, on the information extracted from the beginning (more similar to one image from the choice pair), or rather from the end (more similar to
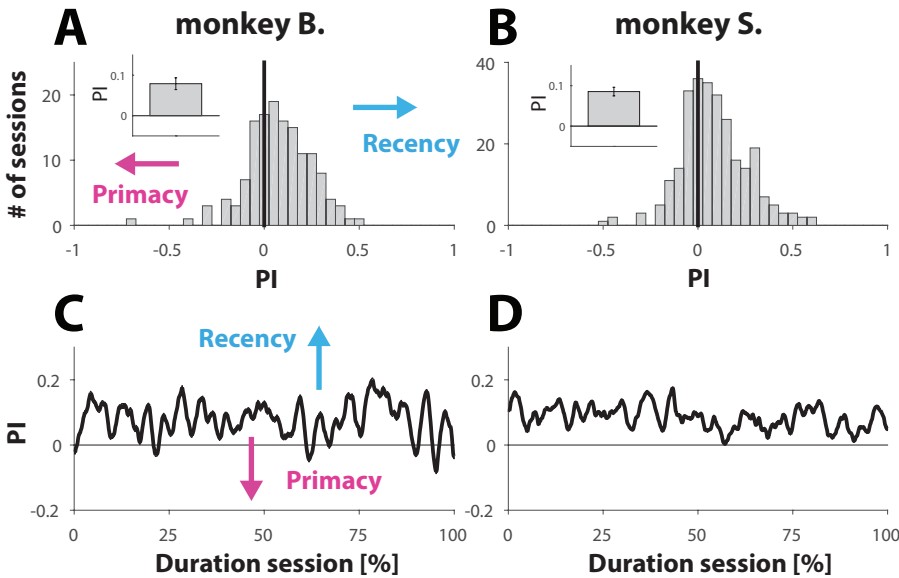

**Figure 3** **Analysis behavior for trials with ambiguous dynamic cues.** (A and B) Histrogram showing the distribution of PI values computed for each session (see Materials and Methods). A PI of zero (marked with vertical line) represent sessions where behavior was equally balanced between 'recency' and 'primacy'. Values are further shifted away from zero (negative for primacy and positive for recency) with bias strength. The insets show the PI value, averaged across all sessions for each primate; error bars denote SE. (C and D). Behavior for trials with ambiguous dynamic cue as a function of normalized trial number. Each session was normalized with respect to trial number. C and D show the average across all normalized sessions.

the other image) of the dynamic cue? To quantify the weight of each of those effects, we calculated a Preference Index (PI; see Eq. (1) in Materials and Methods). This metric yields a value between $-1$ and $+1$ for each session. A value smaller than 0 reveals that for this session the primate was more likely to engage in primacy bias, while a value greater than 0 indicates that the recency effect prevailed. It should be noticed that the terms "primacy" and "recency", for this study, are not used in exactly the same way as for serial position effect studies (*Murdock & Bennet, 1962*), where they refer to the tendency of an individual to respectively best recall the first and last items of a list.

Because weighting of the stimuli had no significant effect on the PI values, we merged both conditions for further analysis. Computation of the mean PI value, averaged across all 141 sessions from monkey B. and all 269 sessions from monkey S. revealed that both primates engaged in behavior that slightly, but consistently, tilted towards recency bias (Figs. 3A and 3B). The averaged PI computed for monkey B. was $0.08 \pm 0.01$ (inset Fig. 3A) and $0.09 \pm 0.01$ for monkey S. (inset Fig. 3B). Although the effect size appears small, it was statistically very significant for each animal, as demonstrated by using a non-parametric sign test yielding p-values of $1.7 \times 10^{-6}$ and $3.2 \times 10^{-9}$ for monkeys B. and S. respectively. This tendency towards recency is also illustrated by the distribution of the PI values (Figs. 3A and 3B), where the bin with value 0 (marked by a vertical black line) represents sessions where the recency and primacy effects were equally balanced.

It is possible that towards the end of the session NHPs, with respect to ambiguous trials, would try "less hard" to find the "correct answer". If so, then the PI value would converge towards 0 over the course of a session, consistent with random and effortless response selection. Taking the same approach as we did to study behavior with respect to unambiguous trials over the course of a session (Figs. 2C and 2D), we also computed the PI value as a function of normalized trial number. The results (Figs. 3C and 3D) suggest that also in this case, the behavior of the NHPs did not change across the duration of a session.

Taken together, the results shown in Fig. 3 suggest that the NHPs, despite receiving reward at random, did not choose randomly but used a choice strategy as illustrated by a small but very significant recency effect.

## The effect of stimulus familiarity on behavior bias

To study the effect of familiarity of the visual stimuli, we examined whether behavior exhibited during a session where a given pair of images was novel (denoted S1) shifted when those same stimuli were used again, later, during a second session (denoted S2). Across both NHPs, we identified 163 instances where the same stimuli were used for two different sessions by the same animal (2 $\times$ 49 sessions from monkey B. and 2 $\times$ 114 sessions from monkey S.). The averaged PI value of all S1 sessions (Figs. 4A and 4B) was positive for both animals ($0.008 \pm 0.024$ and $0.062 \pm 0.015$ for monkeys B. and S. respectively), consistent with recency bias. This was also the case for all S2 sessions ($0.130 \pm 0.023$ and $0.099 \pm 0.017$ for monkeys B. and S. respectively). A paired $t$-test between the PI values of S1 and S2 from each pair, however, revealed that this recency bias strengthened significantly with stimulus familiarity, for both monkey B. ($p = 0.0001$), and monkey S. ($p = 0.041$). This effect of familiarity is further visualized in Fig. 4B, where each of 163 data points (green for monkey B and light purple for monkey S.) indicate a pair of sessions (S1 and S2) where the same stimuli were used. The coordinates of each data point correspond with the PI values of S1 and S2.

Those results are counterintuitive, because it was expected that the NHPs, after gaining experience with the stimuli, would begin to unmask the dubious nature of the ambiguous trials, and start selecting the "target" more randomly, resulting in a PI value closer to 0 (the value associated with chance). Instead, reusing the same stimuli strengthened the recency effect, suggesting that the primates, instinctively, increased the use of a strategy that has not proven to pay off.

This observation gives rise to a new question: "is the lag time between a pair of sessions (S1 and S2) critical for the observed phenomenon?"

To answer that question, we computed $\Delta_{\text{PI}}$ (see Materials and Methods for details), between any pair of sessions where a given set of stimuli was used for respectively the first (S1) and second time (S2). This yielded a single value for each of the 163 pair of sessions revealing whether familiarity with a given set of stimuli would not affect behavior (values close to 0), would induce a shift towards primacy bias (negative value), or cause a shift towards recency bias (positive value).

To take the elapsed time (measured in days) between S1 and S2 into account, we divided the 163 pairs of sessions into 3 bins (Fig. 4D). The first bin contained all the pairs of

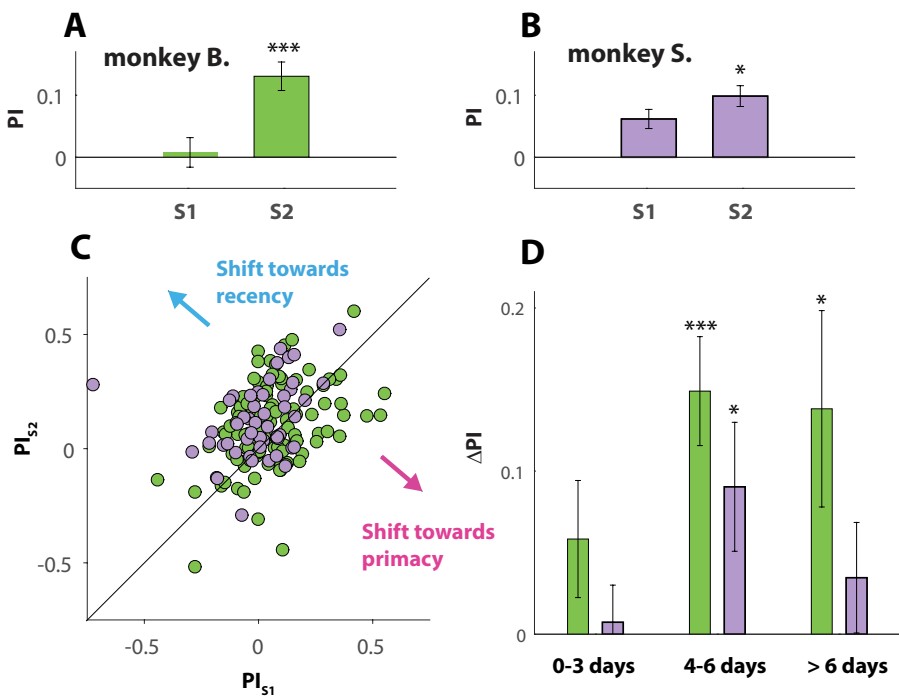

**Figure 4** **Effect of stimulus familiarity.** (A and B) Average PI value (see Matierials and Methods) averaged across all S1 sessions (visual stimuli are novel) and all S2 sessions (visual stimuli are familiar). A PI value of zero is marked with a horizontal line. Only sessions where an S1–S2 pair (matching stimuli) could be formed are included in the analysis. Throughout all panels the colors green and purple are used to indicate data and results respectively from monkeys B. and S. (C) Scatterplot where each circle represents one S1–S2 pair, with $X$ and $Y$ coordinates corresponding to the PI values respectively computed for the S1 and S2 session of that pair. Datapoints on top or close to the diagonal line indicate pairs of sessions where familiarity with the stimuli had little or no effect. Note that most datapoints fall into the upper triangle, consistent with a shift towards greater recency bias with stimulus familiarity. (D) Averaged $\Delta_{PI}$ (see Matierials and Methods) across all S1–S2 pairs, split into 3 bins according to time elapsed between S1 and S2. Note that $\Delta_{PI}$ ranges from −1 to 1, but that all values shown in D are above zero. Error bars in A, B and D denote SE. Asterisks (⋆) indicate statistical significance between S1 and S2 (one, two or three asteriks indicate $p$-values that are respectively smaller than 0.05, 0.01, and 0.001).

sessions where S1 and S2 were separated by 3 days or less (12 pairs for monkey B; 48 pairs for monkey S.); the second bin all pairs of sessions separated by 4 to 6 days (18 pairs for monkey B. and 28 pairs for monkey S.); and a third bin for sessions separated by more than 6 days (19 pairs for monkey B. and 38 pairs for monkey S.). Interestingly, a paired $t$-test between the S1 and S2 sessions revealed that the behavioral shift is not statistically significant (for neither NHP) for the pairs grouped in the first bin, but significant (monkey S.; $p = 0.030$) or even extremely significant (monkey B.; $p = 0.0003$) for the pairs that make up the second bin. The effect then seems to fade away again for sessions separated for more than 6 days (3rd bin), still significant in monkey B. ($p = 0.034$) but not in monkey S. ($p = 0.313$).

If the primates over the course of many sessions, recorded over a period of 10 months, exhibited increasingly more recency behavior, then the observed behavioral shift (Figs. 4A–4C) could simply be explained due to the fact that S2 was recorded at a later date than S1, and not because of familiarity with the stimuli. Plots of PI versus recording day (Figs. 5A and 5B), however, show that the animals did not change their behavior over the course of ~300 days of data collection. The slope of the linear regressions (black lines in Figs. 5A and 5B) are not statistically different from zero ($p = 0.09$ for monkey B., and $p = 0.96$ for monkey S.).

Because those unambiguous trials were rewarded at random, it is not impossible that for some sessions, a given strategy (for instance recency) was rewarded more often, or less often, than what would be expected by chance. To investigate whether our results could be explained by a reinforcement history, we plotted the behavior shift between S1 and S2 ($\Delta$PI) as a function of how often a given strategy, primacy (Fig. 5C) or recency (Fig. 5D), was rewarded during performance of S1. The insets of each figure shows how the linear regression of the data would look if there was a strong effect of reinforcement history. For instance, always (never) receiving reward for choosing recency during S1 would trigger the animal to rely more (less) on recency during S2, which would then result in a more positive (negative) $\Delta$PI (inset Figs. 5D). The analysis (5C and 5D), however, demonstrates that our results cannot be explained by reinforcement history: the slopes of the linear regressions (green line for monkey B. and purple line for monkey S.) in Fig. 5C are not statistically different from zero ($p = 0.55$ for monkey B., and $p = 0.27$ for monkey S.); this is also the case for the regressions shown in Fig. 5D ($p = 0.19$ for monkey B., and $p = 0.92$ for monkey S.)

Taken together, the results, shown in Figs. 4 and 5, suggest that the cognitive processes that took place while the visual stimuli were used for the first time, altered behavior during a later session when those same stimuli were used again. Interestingly, the influence appears to be mild when familiarity with the stimuli is only established recently, but strong when stimuli used for the first time are used again after 4 to 6 days. Unsurprisingly, the effect decreased with longer time between both sessions.

## DISCUSSION

When NHPs, trained to interpret a visual cue, were faced with cues both dynamic and ambiguous, they relied more on the information contained at the end than at the beginning of the cue. Using nomenclature borrowed from cognitive memory studies, we labeled this behavior recency bias. Although our experimental setup differs greatly from those used for serial position studies, the observed recency bias is consistent with that reported for a variety of serial position studies under a variety of conditions (*Bonk & Healy, 2010*; *Bonnani, 2007*; *Farrand, Parmentier & Jones, 2001*; *Tremblay et al., 2006*).

Because those results are neither unexpected nor surprising, and in agreement with the literature they, a fortiori, validate the next level of analysis resulting in our most intriguing finding; i.e., that the observed cognitive bias is strengthened by the familiarity with the stimuli.

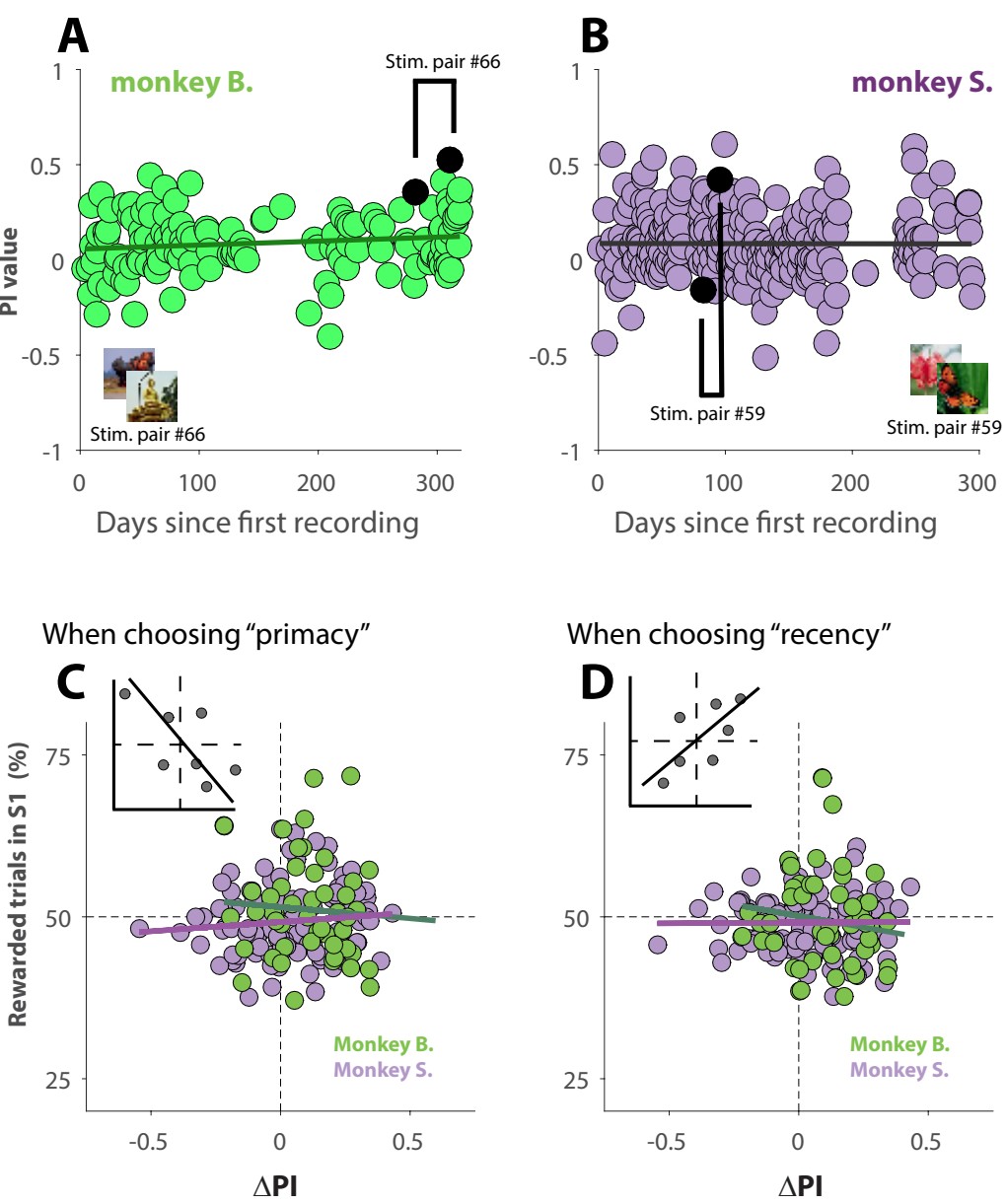

**Figure 5** **Changes in behavior across sessions and effect of reinforcement history.** (A and B) PI values as a function of recording day. Sessions are thus in chronological order. The black lines represent linear regression lines. For illustration purpose, one pair of S1 and S2 was marked in each panel. The parent stimuli for those sessions are shown as insets. (C and D) $\Delta_{PI}$ as a function of reward frequency during performance of S1, for trials where the primates selected the primacy option (C) or the recency option (D). Green (monkey B.) and purple (Monkey S.) lines represent the regression slopes for each dataset. The insets illustrate the expected correlation for behavior affected by a strong effect of reinforcement history.

Non-human primates quickly become experts (*Feng et al., 2009*) in minimizing effort while optimizing reward, for tasks they have been trained in during weeks or months. We were, however, not concerned that the primates would stop making a mental effort by selecting a target at random in response to the "impossible" trials. This because the animals

were trained under conditions that lead to high-probability reward, encouraging them to figure out the rule, or associative contingencies, that leads to reward. The results show, indeed, that the primates—for the "impossible" trials—did not choose at random, but genuinely selected an option anticipated to leading to reward. The results demonstrate that the primates, when choosing one of two possible alternatives are more likely influenced by the information provided at the end of an ambiguous dynamic cue (recency), than at the beginning (primacy).

From a behavioral perspective, the robust increase with stimulus familiarity in recency bias is highly counterintuitive: it confirms that the NHPs—despite having rehearsed with those same stimuli before—not only failed to uncover that the reward for unambiguous trials was non-differential, but relied even more upon a strategy that does not yield benefits.

One obvious explanation for those results is that one of the two parent stimuli was more appealing to the primate. This explanation, however, needs to be discarded because ambiguous dynamic cues were controlled for direction so that choices influenced by a "favorite" stimulus would cancel each other out with respect to primacy and/or recency bias.

More than 70 years ago, Skinner demonstrated that pigeons are susceptible to superstition, by showing that they behave as if there were a causal relationship between behavior and reward (*Skinner, 1948*). Although we cannot prove or disprove that superstitious behavior explains our results, we do not think that this is the case. First of all, recency behavior did not strengthen across sessions (Figs. 5A, 5B); and second, the lack of correlation between how trials were rewarded during S1 and behavior during S2 (Figs. 5C, 5D) makes it unlikely that the animals adapted a behavior reminiscent of superstition.

A more plausible explanation for the observed familiarity effect is representational momentum (*Nijhawan, 1994*), an error in visual perception where an observer believes that a moving object is further along its pathway than it is in reality. In humans, the error is thought to be caused by two factors: the delay in visual processing (*Lappe, Krekelberg & Lappe, 2000*) and intrinsic knowledge of the laws of physics. For instance, if we follow a car speeding along the highway, with our eyes, then our brain will compensate (too much) for the delay in visual processing by anticipating where the car will be instances later. Interestingly, this perceptual error has not only been demonstrated for events hardwired in the perceptual system of our brain (*Shepard, 1984*), but also for dynamic scenes where the chain of events is expected. It is thus possible that this bias can be artificially induced, for instance, by repeated exposure to dynamic images whereby one object is turned into another. After frequent exposure, such an artificial sequence might be perceived as a "natural" chain of events. An older study demonstrated that people had greater difficulty to discriminate between two frames if they were not used to seeing those in sequence (*Freyd, 1983*). A chain of events implies movement, and where there is movement, representational momentum effects might be applicable. Monkeys cannot rationalize what they see as humans do, and therefore might incorporate the morphed directional dynamic images as a physical and natural transformation. If so, then representational momentum might explain the reported familiarity effect. In other words, after having been familiarized with

the stimuli during a previous session, the NHP brain can now anticipate the "next" frame or frames (to compensate for the visual delay) during the presentation of the ambiguous dynamic cue, even if the "next" frame will not be displayed. This anticipation effect might thus weight visual perception in favor of "recency", and explain why we observed an increase of recency bias with familiarity with the stimuli.

Changes at the molecular level induced by novelty or familiarity with visual stimuli have been reported for rats (*Zhu, Brown & Aggleton, 1995*) and mice (*Kissinger et al., 2018*). Other groups observed an increase in evoked potential with familiarity of the visual stimulus (*Anderson et al., 2008*; *Peissig et al., 2006*); *Kissinger et al. (2018)* also reports an increase of theta oscillations with stimulus familiarity. At the behavior level, stimulus repetition has been observed to improve perception and performance. *Grill-Spector, Henson & Martin (2006)* have reviewed the study of the neuronal mechanisms involved in stimulus repetition. One leading hypothesis, at the system level, is that repetition sharpens neural representation, which results in the increase of neuronal synchronous oscillations (*Brunet et al., 2014*). Whether changes in molecular expression, modulation of visual evoked potentials, and/or changes in neural rhythms triggered by stimulus familiarity leads to the modulation of temporal dynamics, and in turn results in altered visual perception that accounts for our results, requires further investigation.

Our results also show that the behavior shift induced by stimulus familiarity is optimal when a set of stimuli is used again after 4 to 6 days (Fig. 4D). Why it peaks for this particular latency is not clear since the timeline does not correspond with any known neurological or psychological process. One possible explanation is that underlying changes in gene expression, induced by novel stimuli (stimuli presented during session S1) require several days to reach full potential. If so, then those molecular changes would reverse back to baseline as time goes by, consistent with the results shown in Fig. 4D (third bin).

Finally, we noticed that the results of our study show remarkable similarities with a phenomenon known as the Illusory Truth Effect. This bias, well studied in humans, shows that repeated statements are more likely judged as true than unrepeated statements (*Begg, Anas & Farinacci, 1992*; *Dechêne et al., 2010*). The leading hypothesis is that the repetition causes processing fluency, which in turn drives the illusory truth effect (*Fazio et al., 2015*); thus not unlike the facilitation of neuronal processing caused by the repetition of visual stimuli (*Brunet et al., 2014*; *Gotts, Chow & Martin, 2012*). In a study where participants were scanned with fMRI while rating the truth of unknown statements, one brain region, the perirhinal cortex, was shown to interact between repetition and ratings of perceived truth. Interestingly, activity in that area only increased for statements that were presented to the participant before being scanned (and thus repeated) but not for new ones (*Wang et al., 2016*). Intriguingly, rhesus macaques lesioned in this same area (the perirhinal cortex) exhibit familiarity judgment deficits, requiring more exposure to objects before they are able to judge them as familiar compared to control animals (*Weiss et al., 2017*). In both cases (illusory truth effect and the here reported behavioral shift) the subject is unaware that repeated exposure leads to the strengthening of an unproven concept (reliability of statements in humans, and decision strategy in monkeys). Whether those apparently different biases, share common neural correlates appeals for further investigation. The

illusory truth effect explains our propensity to accept fake news as true (*Lazer et al., 2018*), which has been called a threat to democracy by Barack Obama in 2016. An animal model that mimics this cognitive bias could thus fuel future studies that address this bias that has such a far-reaching implication.

## CONCLUSION

This study reports the following findings: (1) NHPs can be trained to excel in a 2AFC task that features static as well as dynamic cues. (2) NHPs do not choose haphazardly when faced with trials where the visual cue is ambiguous and the reward given at random; they instead use a choice strategy that differs from non-random behavior. For those "impossible trials", the NHPs based their response on the information provided at the end of the dynamic cue (recency) rather than at the beginning of the cue (primacy). The observed recency bias was small but consistent and very significant. (3) Interestingly, the recency bias was stronger for a session where a set of stimuli was used a second time compared with that of the session where those stimuli were used for the first time. (4) The magnitude of the behavior shift caused by stimulus familiarity seems to depend upon the time elapsed between S1 and S2 (Fig. 4C). For now, we are limited to speculation in order to explain the results (3) and (4). More study is needed to better understand the cognitive processes involved.

## ACKNOWLEDGEMENTS

The contents of this article are solely the responsibility of the authors and do not necessarily represent the official views of the NIH.

### Funding

Nicolas Brunet received grant support by an Institutional Development Award (IDeA) from the National Institute of General Medical Sciences of the National Institutes of Health (INBRE, P20GM103476). The funders had no role in study design, data collection and analysis, decision to publish, or preparation of the manuscript.

### Grant Disclosures

The following grant information was disclosed by the authors:
National Institutes of Health, National Institute of General Medical Sciences: INBRE P20GM103476.

### Competing Interests

The authors declare there are no competing interests.

### Author Contributions

- Nicolas Brunet conceived and designed the experiments, performed the experiments, analyzed the data, prepared figures and/or tables, authored or reviewed drafts of the paper, approved the final draft.

- Bharathi Jagadeesh conceived and designed the experiments, contributed reagents/-materials/analysis tools, authored or reviewed drafts of the paper, approved the final draft.

## Animal Ethics

The following information was supplied relating to ethical approvals (i.e., approving body and any reference numbers):

All animal handling, care and surgical procedures were performed in accordance with guidelines established by the NIH, and approved by the institutional Animal Care and Use Committee (IACUC) at the University of Washington (3275-01).

## Data Availability

The raw data is available at Figshare: Brunet, Nicolas (2019): Familiarity with visual stimuli boosts recency bias in macaques.. figshare. Dataset. https://doi.org/10.6084/m9.figshare.8251523.

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
