# Peer review of "Familiarity with visual stimuli boosts recency bias in macaques"

_PeerJ, doi:10.7717/peerj.8105_

## Round 0.1 · original submission · Major Revisions

Based on your Appeal and stated confidence in being able to address the reviewers' and my concerns about the validity of your methods to test your research question, I am happy to invite you to prepare a major revision., instead of the prior Rejection decsion

· Appeal

Appeal


· · Academic Editor

Reject

I have been fortunate to have two experts review your MS. Whereas both find the data interesting and worthy of exploring further, neither are convinced by your interpretation. Thus, they suggest substantial reframing and additional work to clarify the nature of the contribution. I concur with these reviewers that there is something of interest here but I believe you would need to write a different paper, with additional data, in order to make a clear contribution to the literature. I agree with the reviewers that you have neglected alternative explanations. Further, there is a literature on primates' responses to ambiguous stimuli within the ambiguous cue paradigm and the cognitive bias literatures that could be tapped into. Thus, unfortunately, I must reject the paper in its current form.

Reviewer 1 ·

Basic reporting

Overall, I was able to follow this manuscript without great difficulty. I think the presentation of some of the material could be reordered to improve the flow in some instances and there are a few other minor suggestions for the author:
1. A few minor typos throughout (e.g., line 14 has "despites" when "despite" is grammatically correct).
2. There are some anthropomorphisms throughout that imply psychological mechanisms involved that I do not think are called for (e.g., line 15 has "animals were further deceived into believing..." - Animal behavior is not always determined be beliefs, but can be the result of lower level heuristics such as bias or habits that do not come from incorrect judgments).
3. Lines 37-41: From the reading of this section I am not sure what the "fluency of thought" hypothesis is or how it was supported by the study discussed.
4. Line 69: "behavior response" is redundant.
5. Some of the text in the Results section belongs in the Methods section or is redundant with it.
6. Line 137: A missing reference is flagged - "(ref)".
7. The logic behind the use of "weighted" cues were never clearly described (from what I could find). I have no idea on how to interpret the absence of its effect because I don't know why it was ever included.
8. Line 198: "The impact of this apparently considerable modification of behavior was however very modest at best." This sentence seems contradictory in that the impact was considerable and modest.
9. More elaboration (or reference if it is commonly used elsewhere) is needed for the Behavioral Shift model for me to understand what is going on (Lines 127-133; 292-297).
10. Figure 1 was excellently designed to help follow the procedure.
11. Line 234: The behavior was slightly above chance (~53-55%) so I wouldn't say that it was "dominated" by the recency effect.
12. The caption for Figure 4 needs clarification. I had a difficult time following what was plotted (and why).

Experimental design

I have some questions about the subjects and apparatus: What were their ages? What was the reward? Was the animal restricted of the reward prior to the session to maximize motivation?

Line 95: Blocks are first mentioned without explanation of what they were, how they were arranged, and why they are part of the design. This is a procedural detail I am not sure I could replicate. Also, how many trials per session (or other session duration criteria)?

Lines 171-173: A claim for no effect of the play direction of the cue is made without supporting statistics (even though at a glance it appears it would be the case). Similar issues with clarity of statistical reporting (e.g., why weren't stats reported and in some cases what statistics were used and what were the comparisons) occurs in other places (e.g., Lines 198-205; 277-279; 308-309).

Validity of the findings

The claims that the data support the repetition bias effect in humans does not rule out alternative interpretations - such as responding to familiar stimuli that produced reward in the past (albeit, intermittently). Linked to my criticism of the anthropomorphism about their "belief," these choice data could be a product of another mechanism and more work is necessary to make the claim that these choices were driven by some form of conscious belief. I'm not immediately sure what could, but assessments like NHP meta-cognition would likely make the strongest case for that position (e.g., Middlebrooks & Sommer, 2011; JD Smith, 2009, etc.). These data might be similar to the Illusory Truth Effect, to the extent that both involve responding to stimuli that are relatively more familiar (even if vaguely so), but we don't have a strong reason to believe that the monkeys would believe those stimuli to be more "truly correct" in a (metacognitive) confidence judgment task (however, that might be a very interesting experiment).

The primacy and recency effects is often involved in recalling stimuli that are presented in a list and individuals having a better time remembering the first and last items compared to the middle. This task with the monkeys doesn't systematically test their memory, (i.e., they don't get tested on all morphed stimuli in the series and show better responsiveness to the first or last times), it just demonstrates that they are (a little) more responsive to what they most recently saw. This could be a recency-like effect, but it might be that when rewards are random that they simply are slightly more likely to just select what best resembles what they just recently saw. Since the monkeys demonstrated good judgment when the unambiguous cues were presented forward and backward, it would seem that they have great memory in this task.

The outcome of the behavioral shift (where the recency effect is strongest in intermediate session lags, but not the most recent) is unusual and needs more discussion. This latency to showing a large recency effect might be a clue in what neuro/psychological processes are involved. Also, in Figure 4, why doesn't the BS show both monkeys separately? Was the effect shown representative of both monkeys?

Additional comments

I find these data interesting and would like to see them published. That monkeys could attend to and discriminate between cues that are animated is novel (as far as I know) and could make for a nice small report.

I do, however, have issues with how the data were interpreted that either needs a stronger defense or needs adjusting (see Validity of the findings). There are a few instances where I would not be able to directly replicate what was done in this study and requires more clarity and better organization (see Experimental design). I would require these issues be addressed in some way before I could recommend publication. Beyond that, I think there are some stylistic changes that would make reading this manuscript easier (see Basic reporting).

Reviewer 2 ·

Basic reporting

The manuscript would be improved by careful editing and reorganization, eliminating redundancies. The authors' meaning is generally clear, but there are places (noted below) where clarity can be improved.

Given the authors' interpretation of their effects, the literature cited is largely drawn from studies of illusory truth effects in humans.

The structure of the article is appropriate, but some of the figures add little to what is reported in the text.

Lines 34-36: Indeed this is true, but the same effects may have affected other (and earlier) elections as well, and countless other decisions. “Fake news” may be a recent term, but decision biases due to repetition certainly are not.

Lines 39-41: additional details are needed to tie the evidence of perirhinal activity and fluency of thought

Line 44: change to “superior spatial and temporal resolution….that are required…”

Lines 43-45: This is confusing. The techniques used in the study under consideration are neither noninvasive, nor high resolution

Line 51: change to “two-alternative forced choice (2AFC) task…”. Note that this is actually a 2AFC DMTS task. (A 2AFC task could also be two-choice discrimination learning, or two-alternative matching-to-sample, or similar paradigms.) I suspect that the authors are using 2AFC rather than DMTS because, on ambiguous trials, there is no direct match-to-sample.

Lines 51-63 require careful editing and revision. It is difficult to understand what the authors are trying to say. Given that this section is largely redundant with the methods, these lines can be deleted to improve clarity.

Line 66: change to “Data were obtained….” Also Line 234 to “…data unmistakably show…”

Lines 76-77: Clarify: all animals and trials used one single pair of images that was randomly selected, or each trial used a pair of images randomly selected?

Line 137: add citations at “ref”

Lines 136-140: This paragraph would be more appropriate in the Method section. It certainly is not Results, unless the authors report the training curves for the DMTS task. Lines 141-156 are also largely redundant with the Method. Reorganize to eliminate repetition, and report results in the Results section.

Line 165: What does “data and results not shown” mean?

Line 166: Use “while” and “since” only to denote the passage of time; in this sentence, change “While” to “Whereas”

Line 167, “as explained later”: These trials were explained already, in the method section

Figures 2A and 2B are not very informative, given the scale of the Y axis

Lines 178-191: This is Discussion, rather than Results

Line 183: change to “This is because…”

Line 191: What constitutes a success on these ambiguous trials? Do you just mean the percentage of trials that were rewarded? This was set at 50% by design. There was no strategy to be figured out.

Lines 199-203: It isn’t clear why this condition was introduced. This should be explained in the Method section. If the means are not significantly different from the continuous dynamic morphs, then there is no need to speculate on what caused the difference (i.e., there isn’t one). If the means are significantly different, report the statistics to show this and then interpret. Also, I don’t understand the last sentence of this section.

Line 208: Here and in many other places, the authors use a semicolon with no conjunction. A period to form separate sentences is more appropriate and easier to read.

Lines 208-209: Unnecessary explanation of how to compute a percentage

Line 226: revise to “…one image from…”

Lines 227-228: Yes, these terms were borrowed from the study of serial-position effects, where they mean something quite different than what is meant here

Lines 255-259: This result is unsurprising. The recency image was closest to the choice images temporally. Further, it might be noted that, in investigations that produce serial position curves by animals, that recency biases are common, as the authors acknowledge in the Discussion.

Lines 265-269: Redundant with Method section

Experimental design

This is original primary research within the scope of the journal. The method (testing the effects of repetition on the categorization of ambiguous dynamic morph stimuli) is tangentially related at best to the human illusory-truth paradigm. Analyses are unclear and seem likely to violate assumptions of the inferential statistics employed.

Lines 168 and 173: Where are the inferential statistics to support these claims?

Lines 199-203: It isn’t clear why this condition was introduced. This should be explained in the Method section. If the means are not significantly different from the continuous dynamic morphs, then there is no need to speculate on what caused the difference (i.e., there isn’t one). If the means are significantly different, report the statistics to show this and then interpret. Also, I don’t understand the last sentence of this section.

Lines 234-243: It is not clear how these t-tests were conducted. If session-by-session comparisons of PI and RI were made, separately for each animal, then the test violates the assumption of independence; that is, RI=100-PI. A nonparametric statistic (or simple z-test for comparing significance of proportions) should be used. In any case, 55% vs 45% hardly seems dominant. Further, inferential statistics cannot yield a p value of 0.

Lines 311-316: What is the relation between the animals’ choices on S1 (rewarded compared to not) and their choice on S2? The question is about specific stimuli: Were the monkeys more likely to repeat a response to an ambiguous stimulus in S2 if they had been rewarded for that response in S1.

Validity of the findings

This is an interesting and novel study, and produces a reliable effect that requires explanation; however, I am highly skeptical of the interpretation that the authors advanced. Rather, it seems to me that the bias toward ‘how the stimulus ends’ rather than ‘how the stimulus begins’ is predicted by the literature on recency effects proper, and the increase in this bias with repetition is interesting but inconclusive with respect to what the animals “believed” about the stimuli.

Line 61: does this study actually reveal anything about beliefs?

Lines 288-289: This study reveals nothing about the animals’ beliefs. The animals may have come to believe that their strategy was correct, as the authors claim. The animals may have come to understand that strategy didn’t matter, but there is no reason to assume that responding would have approximated chance in that instance. If strategy didn’t matter, the animals may have perseverated on the ‘recency’ strategy (as indeed the data suggest). The results are ambiguous with respect to what the animals were thinking. Because the majority of responses were to the recent image, and responses were randomly rewarded, the majority of rewards on ambiguous stimuli followed RI responses.

Line 312, “…strongly affect”: this seems over-stated for changes in response of this magnitude; change to “…significantly affect…” (although I have concerns about the appropriateness of using t-tests to compare RI and PI).

Lines 338-343: An increase in salience of the recency portion of dynamic stimuli, or at least the increased control over behavior of this portion of ambiguous dynamic stimuli, was observed. This increase was observed for stimuli that repeat (S1 and S2), and is not predicted by trial-to-trial effect. However, none of this suggests anything definitive about the animals’ beliefs. Shifts in responding away from randomness can indicate learned helplessness as well as learning a response strategy. It is certainly a bold leap to conclude that the finding is functionally related to the effects of repetition on illusory truth.

Additional comments

I commend the authors for attempting to extend to nonhuman primates a phenomenon heretofore reported only for humans. Additional analyses (i.e., of the response on S2 conditioned on the response and outcome of S1, to test for stimulus-specific memory effects) and other edits (e.g., careful editing to improve organization and clarity and to eliminate redundancy) are recommended as the authors revise the manuscript for future submission.

---

## Round 0.2 · Major Revisions

Thank you very much for taking the reviewers’ comments seriously and making substantial revisions to your manuscript. I appreciate that you have reframed the data. However, I agree with Reviewer 2 that it would be prudent to change the title of your MS accordingly, given that you no longer focus on the illusory truth effect. Reviewer 1 feels that your revisions have satisfied most of their concerns, although they still make a number of valid points that should be considered. Reviewer 2 is more favorable to the MS this round but still has fairly significant concerns. At the very least, you need to include the requested analysis on stimulus specific effects. Thus, I would encourage you to engage in another revision.
I have a few comments of my own.
The ambiguous cue literature should be discussed. I don’t follow the discussion on lines 52-61. There are no references so it is not clear what paradigm or previous studies are being referred to here. It sounds like the authors are blending components of judgement bias and ambiguous cue tasks without accurately describing either. No results are discussed from previous studies. It would be difficult for a naïve reader to follow the point being made here.
There is no real rationale for the significance of examining the effects of familiarity in this task. Why is it important to consider how familiarity factors into decision biases?
I am not sure why the primate are described as being “deceived.” After reading beyond the abstract, it is clear that this refers to ambiguous trials that do not contain useful information. Thus, use of the term deception might be consistent with passive deception, but most readers will anticipate an active attempt to deceive, which is not tested here.
The subjects section should focus only on describing the subjects and include information about age, rearing history, experimental history and where they are housed. Methods should indicate the criteria for passing the training phase and clearly differentiate training from testing.
There is still much missing detail in the Methods. What software was used to program and present the stimuli? Why so many more sessions for Monkey B compared to Monkey S?
Did you collect data on response times? It seems that effortless responding on ambiguous trials could be assessed by how quickly they responded.
On line 15 in the abstract, task should not be pluralized.
On line 34, consequences should not be pluralized or get rid of the “a” on line 33. Please watch for similar singular/plural disagreement throughout.
In line 18 (and check elsewhere), please avoid valence terms such as “performed excellent.” It is neither grammatical nor appropriate. “Performed at high levels of accuracy” would be preferable.
I still don’t really agree with the interpretation of results. Just because the monkeys did not adopt a non-random strategy, it does not imply they were engaging in effort to find the correct answer. Primates often adopt consistent strategies like side biases, but may not indicate effortful attempts to engage with the task. It may be less effortful to adopt consistent strategies.
Delete do on pg. 55.
On line 142, change “effect” to “affect.”
On line 156, change “analysis” to “analyses.”
On line 160, do not pluralize “metrics.”
On line 227, consistently is misspelled.
On line 229, statistically is misspelled.
Reword line 301.
Use commas after i.e., or e.g.,
Please check the revised MS carefully for typos and grammatical errors.

Reviewer 1 ·

Basic reporting

The introduction does a better job setting up the the experiment and I like that the predictions were explicitly spelt out. I still think the intro could be fleshed out a little better to aid the naive reader (e.g., smoother transitions, more explicit clarity of what the predictions are for the current study), but I don't feel such adjustments are critical.

Some phrasing could be adjusted for clarity of meaning: e.g., Line 201: "we were hopeful that the monkeys would make a genuine attempt to identify the “correct” target..." could be rephrased as "We believe that when the ambiguous cues were presented that the monkeys were responding based upon the learned rules for making a "correct" response for the dynamic trials, rather than responding randomly to trials where the underlying rule made reward delivery random"). A minor quibble, but everyday terms like "genuine" has implications that the monkeys could and would make "disingenuous" responses. That said, if the meaning of that phrase was made very clear, I wouldn't oppose this phrasing in a discussion section where (I, at least, would allow) authors a little more artistic leeway, but I don't like it in the methods.

A few minor typographical errors linger (e.g., Lines 39, 86, 143, 179, 217, 319, 327)

Experimental design

No Comment

Validity of the findings

I do not quite follow the "representational momentum" explanation for the data (Lines 326-343). I don't have a background in that literature and so I think further clarification might be helpful for naive readers to 'connect the dots' linking this effect and the effect in the paper.

I think the discussion of the effects of stimulus repetition is a much stronger candidate explanation for these results. It, however, could be tied into the results a little better in terms of learning (e.g., if the animals were biased to adopt a 'recency' strategy for the ambiguous trials, then repeating those trials with familiar stimuli may strengthened that strategy even if that strategy was irrelevant to the outcome). This effect might be somewhat related to "superstition" in nonhuman animals (e.g., Skinner, 1948), this however is an off-the-cuff speculation that would need a more thorough consideration.

I now like how the Illusory Truth Effect is integrated into the paper, it fits better given the discussion of repetition the outcomes in this paper.

Additional comments

Overall, this revision is much improved over the original and appears to address the major points that I brought up. I do have some more minor suggestions for improvement, but nothing must be necessarily addressed to prohibit publication. My best wishes to the authors.

Reviewer 2 ·

Basic reporting

The manuscript is improved tremendously over the original submission and, although editorial editing is still required, the manuscript is easy to read and understand. Extensive revisions have addressed most of the reviewers’ issues with the original manuscript. Specifically, my concerns about the focus of the literature review, the heavy discussion of what the animals supposedly believed and its relation to the illusory truth effect in humans, and the statistical analyses of the original manuscript have been ameliorated in the revision.

Lines 38-42: Everything said here is accurate; however, this memory effect is not a neuroeconomic / decision-making outcome (other than in the broad sense that the monkeys have to decide how to respond, under which umbrella every study is arguably a decision-making study). I think the authors are just trying to show how biases from research with humans have been replicated with nonhuman animals. An example from the decision-making literature (e.g., studies of Monty Hall performance or even visual illusions with animals) would be better.

Lines 310-311: If only the monkeys understood that these trials were being nondifferentially rewarded! Certainly monkeys will try to maximize rewards and minimize effort. But these are animals that have been trained under conditions that lead to high-probability reward. They don’t know that reinforcement is random, and thus are likely trying to figure out the rule or associative contingencies to earn reward on every trial. They may continue to try strategies (even those that aren’t fruitful or efficient, as when an animal develops a superstitious behavior).

The figures are interesting and informative, although their scale underscores the fact that these are small effects (e.g., each line in F2c,d is about 5% thick; each dot in F4c covers about 0.1 of each axis).

Experimental design

This is original primary scholarship that falls within the scope of this journal. The use of nondifferentially reinforced ambiguous stimuli to probe for perceptual or response biases is a standard and successful procedure (e.g., Parrish, Brosnan & Beran’s 2015 study of the Delboeuf illusion in humans and monkeys). The morphed stimuli used here are novel and interesting. The experimental procedures included appropriate controls.

Examination of Figure 1b indicates a potential issue that I had failed to notice in my original review. Although the probe stimuli are, by definition, ambiguous, as they cross the line between stimulus A and B categories, and include stimulus states (frames 4 and 5) that had been rewarded as part of stimulus A as well as stimulus states (frames 7 and 8) for which the animal had a reinforcement history associated with stimulus B. However, the recency portions were not ambiguous. If the authors are correct that familiarity biases the animal toward recency, then one would expect the same effect had the ambiguous stimuli been the perceptually more ambiguous frames 5,6,7 or 7,6,5. This could be tested in future studies.

Validity of the findings

Although the effect sizes are not large, by the authors’ own admission, it seems undeniable that the monkeys were biased in the direction of the most recent component of the ambiguous stimuli. It seems equally clear that this bias gets stronger when the monkeys are viewing an ambiguous stimulus a second time, even though days intervened between viewings on most occasions.

What is less clear is what this pattern of results means, and whether it is a substantive contribution to the literature. I actually find the most surprising result to be the fact that the recency bias wasn’t even stronger, and didn’t get substantially stronger across training. These results, plus the novelty of the question and methods, might justify publication. But what would help in my interpretation of the change in performance with familiarity is to see the recency bias on S2 as a function of what the response was on S1, and whether that response was rewarded. I strongly suspect—and I don’t think the authors disagree, based on their discussion—that in the absence of another categorization rule, monkeys are biased toward the most recent stimuli, and that the effects reported here can be modeled accurately on the basis of reinforcement history and stimulus generalization (i.e., the most recent image of an ambiguous stimulus has a reinforcement history that is greater than the first image in that ambiguous sequence).

Although the authors are correct that these results strengthen an extant literature on monkeys’ preference for recency vs primacy, this replication alone is not sufficiently substantive to merit publication.

Although the illusory truth discussion is appropriately constrained to one suggestive paragraph, it remains in the title, promising more than the study can deliver.

Additional comments

Congratulations on a greatly improved manuscript, showing careful attention and thoughtful responses to earlier comments. If the analysis I suggested in my original review (i.e., of the response on S2 conditioned on the response and outcome of S1) is added, I might have more confidence in recommending the manuscript for publication.

---

## Round 0.3 · Minor Revisions

Thank you for your continued attention to the reviewers’ concerns and comments. I appreciate the revisions you have made. However, there are a number of sloppy errors that need to be corrected in the revised MS before it can be accepted. See below. The line numbers refer to the MS with changes tracked.

There is something off about lines 64-65. Please correct.

On line 66, delete the “s” on predicts.

Avoid using ; twice in one sentence (e.g., line 69). It is easy to break this up into separate sentences.

Correspondingly is misspelled on line 70.

Delete the . after environments on line 72.

Separation is misspelled on line 73.

There is a trailing “and” on line 74.

Replace “we” on line 84 with “researchers”

On line 89 place “in” between “interested” and “how.” Similarly for line 286.

On line 99, place a space between the reference and the next sentence.

On line 131, delete the I on suggest

Are you missing “not” in front of “rewarded?” on line 142?

Delete the extra a on line 143.

The sentence on lines 145-148 is not a complete sentence.

On line 229, developed is misspelled.

Insert commas after clauses, such as “For each session,…” on line 238. Check throughout.

On line 291, move “only” to after “contained.” Place a c before ‘which.”

On lines 349 and 353, should there be a ‘not” before “change” and “affect behavior?”

Instead of “keep applying” replace with “Continuing to apply” on line 362.

Delete the extra . on line 363.

Proble should be probe on line 363.

Change person to “individual” on line 385.

On line 391, delete the “t” on although.

On line 513, you mean “prove or disprove.”

Delete the extra ) on line 534.

On line 546, “see” should be “seeing.”

On line 544, the author’s name does not have to be repeated.
Why is the entire name, Bonanni capitalized in both the text and reference list?

Given the large number of errors, please reread the entire MS carefully before resubmitting.

Delete reference to “deceptive” on line 424 as discussed in the last round of review.

---

## Round 0.4 · accepted · Accept

Thank you for attending to the errors. Apologies for the mismatching documents.